# Incremental Predictive Value of Longitudinal Axis Strain and Late Gadolinium Enhancement Using Standard CMR Imaging in Patients with Aortic Stenosis

**DOI:** 10.3390/jcm8020165

**Published:** 2019-02-01

**Authors:** Lucia Agoston-Coldea, Kunal Bheecarry, Carmen Cionca, Cristian Petra, Lelia Strimbu, Camelia Ober, Silvia Lupu, Daniela Fodor, Teodora Mocan

**Affiliations:** 12nd Department of Internal Medicine, Iuliu Hatieganu University of Medicine and Pharmacy, 400006 Cluj-Napoca, Romania; kunalvarmab@gmail.com (K.B.); cristian.petra@gmail.com (C.P.); dfodor@umfcluj.com (D.F.); 2Affidea Hiperdia Diagnostic Imaging Center, 400015 Cluj-Napoca, Romania; carmen.cionca@hiperdia.ro; 3Niculae Stancioiu Heart Institute, 400001 Cluj-Napoca, Romania; lstrimbu@yahoo.com (L.S.); cami.ober@yahoo.com (C.O.); 45th Department of Internal Medicine, University of Medicine and Pharmacy of Tirgu Mures, 540139 Tirgu Mures, Romania; sil_lupu@yahoo.com; 5Department of Physiology, Iuliu Hatieganu University of Medicine and Pharmacy, 400006 Cluj-Napoca, Romania; teodora_mocan@yahoo.com

**Keywords:** severe aortic stenosis, longitudinal axis strain, late gadolinium enhancement, cardiac magnetic resonance imaging

## Abstract

To analyse the predictive ability and incremental value of left ventricular longitudinal axis strain (LAS) and late gadolinium enhancement (LGE) using standard cardiovascular magnetic resonance (CMR) imaging for the diagnosis and prognosis of severe aortic stenosis (AS) in patients with an indication for aortic valve replacement. We conducted a prospective study on 52 patients with severe AS and 52 volunteers. The evaluation protocol included standard biochemistry tests, novel biomarkers of myocardial fibrosis, 12-lead electrocardiograms and 24-hour Holter, the 6-minute walk test and extensive echocardiographic and CMR imaging studies. Outcomes were defined as the composite of major cardiovascular events (MACEs). Among AS patients, most (*n* = 17, 77.2%) of those who exhibited LGE at CMR imaging had MACEs during follow-up. Kaplan–Meier curves for event-free survival showed a significantly higher rate of MACEs in patients with LGE (*p* < 0.01) and decreased LAS (*p* < 0.001). In Cox regression analysis, only reduced LAS (hazard ratio 1.33, 95% CI (1.01 to 1.74), *p* < 0.01) and the presence of LGE (hazard ratio 11.3, 95% CI (1.82 to 70.0), *p* < 0.01) were independent predictors for MACEs. The predictive value increased if both LGE and reduced LAS were added to left ventricular ejection fraction (LVEF). None of the biomarkers of increased collagen turnover exhibited any predictive value for MACEs. LAS by CMR is an independent predictor of outcomes in patients with AS and provides incremental value beyond the assessment of LVEF and the presence of LGE.

## 1. Introduction

Aortic stenosis (AS) is currently the most often encountered valvular heart disease in adults. The overall prevalence of this condition has recently increased as a consequence of progressive ageing of the population, reaching a maximum in patient’s ≥75 years of age (2.8% of the general population) [1]. Severe AS is associated with changes in left ventricular (LV) geometry and function induced by increased afterload [2,3]. In the early stages, the constant exposure to elevated afterload leads to progressively increasing LV wall thickness, a compensatory change that temporarily relieves wall stress. At this point, LV systolic function is preserved. In later stages, compensatory mechanisms are overwhelmed by the constant exposure to increased afterload, leading to maladaptive remodelling and impaired LV systolic function [4,5]. Chronic inflammation, osteoblast activation, active valve mineralization, valvular and myocardial fibrosis occur due to the continuous exposure to increased pressure overload and are important factors in the evolution of AS [6]. Myocardial cell apoptosis [7] and the secretion of excess extracellular matrix proteins [8] favour the development of myocardial fibrosis and lead to systolic and diastolic LV dysfunction and increased ventricular stiffness [9]. Such changes are associated with a higher risk of major adverse cardiovascular events (MACEs) [10,11] and poorer outcomes in terms of clinical status and long-term survival after aortic valve replacement [12]. Changes in LV mechanics usually occur before the overt impairment of the LV systolic function, as assessed by the left ventricular ejection fraction (LVEF). Moreover, LVEF has a high inter- and intra-observer variability (14%) [13] and only becomes impaired in the late stages of the disease [14]. By contrast, global longitudinal strain has increased reproducibility, and is altered from the early stages of severe AS [15,16]. Although echocardiography is valuable, it has some limitations, and additional imaging techniques may provide incremental value for diagnostic and prognostic purposes in severe AS patients. Cardiac magnetic resonance (CMR) imaging is widely accepted as the non-invasive modality of choice for visualizing and quantifying myocardial fibrosis by late gadolinium enhancement (LGE) techniques [17]. A mid-wall pattern of fibrosis has been observed in the myocardium of up to 38% of patients with moderate or severe AS and has been associated with a more advanced hypertrophic response [10], and an eight-fold increase in mortality [10].

Recently, left ventricular longitudinal axis strain (LAS) by CMR imaging has been validated as a fast and reliable method for quantifying global LV longitudinal function [18], which does not require additional pulse sequences and off-line processing using dedicated software tools. The aim of the current pilot study was to assess the incremental value of LAS and LGE using standard CMR images for the diagnosis and prognosis of patients with severe AS undergoing aortic valve replacement.

## 2. Material and Methods

### 2.1. Study Patients

We conducted a prospective study on 128 patients with severe AS undergoing aortic valve replacement and who were examined in the 5th Department of Internal Medicine of the Iuliu Hatieganu University of Medicine and Pharmacy, Cluj-Napoca, Romania, between March 2016 and August 2018. Severe AS was defined as (1) peak aortic jet velocity ≥ 4 m/s, and/or (2) mean transvalvular gradient ≥ 40 mmHg, and/or (3) aortic valve area (AVA) ≤ 1.0 cm^2^ [19,20]. 

Patients who had contraindications for CMR (including incompatible metallic devices, significant chronic renal disease with estimated glomerular filtration rate < 30 mL/min/1.73 m^2^, or claustrophobia), other significant valvular disease, rheumatic valve disease with significant (at least moderate) mitral stenosis, post-irradiation AS, history of previous myocardial infarction with or without coronary revascularization by percutaneous coronary intervention and/or bypass, previous surgery for valvular disease, active inflammatory, infectious diseases, or neoplasia, cirrhosis, pulmonary fibrosis, poor echocardiographic window or those who did not agree to participate were excluded. The total number of excluded patients was 76 (Figure 1). Finally, 52 patients with severe AS (test group) were compared to 52 volunteers (control group), matched for age and sex, who responded to a questionnaire excluding any history of chronic heart disease, other than systemic arterial hypertension, without ongoing symptoms, normal clinical examination and normal electrocardiogram, chest radiographs, echocardiography, Holter ECG monitoring for 24 h and CMR imaging. Volunteers in the control group were recruited during the same enrolment interval as test group patients.

The current study was approved by the Ethics Committee of the Iuliu Hatieganu University of Medicine and Pharmacy, Cluj-Napoca—decision number 196/10.03.2016. The research was conducted in compliance with the Declaration of Helsinki. All patients were informed about the investigation protocol and signed a written consent form before being assigned to either the test or control group. Each patient underwent the same investigation protocol, including medical history, clinical examination, the recording of a 12-lead electrocardiogram, 24-h Holter monitoring, 6-minute walk test (6MWT), biochemical analysis, echocardiography and CMR imaging, which were all performed during the same hospital visit. 

### 2.2. Medical History and Clinical Examination

We recorded the medical history and cardiovascular risk factors of all enrolled subjects, including active smoking, systemic arterial hypertension, dyslipidemia, diabetes, and obesity. Coronary artery disease was considered present if myocardial infarction, angina pectoris, or revascularization (by either angioplasty or coronary artery by-pass) were performed. The New York Heart Association (NYHA) functional classification was used to assess the severity of dyspnoea. Patients with severe AS were considered symptomatic if they experienced dyspnoea, angina and/or syncope. A 12-lead electrocardiogram was recorded at enrolment and 24-h Holter monitoring was performed. The 6MWT was performed in the presence of adequately trained medical personnel following the American Thoracic Society guidelines, and the 6-min walk distance (6MWD) was reported. The risk of mortality after cardiac surgery was assessed by the EuroScore II [21] software available on www.euroscore.org.

### 2.3. Biochemical Analysis

Two peripheral venous blood samples were collected at enrolment from each participant and serum was separated by centrifugation. One sample was used for standard biochemistry tests, including high sensitive C reactive protein (hs-CRP) measurements, with a Beckman Coulter AU480 Chemistry Analyzer. The other serum sample was preserved at −80 °C until the end of the study and used for measuring procollagen type I C-terminal propeptide (PICP), procollagen type III N-terminal propeptide (PIIINP) and N-terminal pro-Brain Natriuretic Peptide (NT-proBNP) levels. All biomarker levels were determined by the Sandwich ELISA technique according to the manufacturer’s instructions (Elabscience Biotechnology Co., Ltd., WuHan, China) For determining PICP, we used Human PICP ELISA Kits, Elabscience Biotechnology Co. PIIINP was determined using Human PIIINP ELISA Kits, and for NT-proBNP levels, Elabscience Biotechnology Co., Ltd., kits were used. Measurements were performed by a single investigator who was blinded to all clinical and imaging data. Renal function was estimated by the glomerular filtration rate (eGFR), using the Chronic Kidney Disease Epidemiology Collaboration equation, considering age, race, gender, and plasma creatinine concentration. Renal function was considered impaired at eGFR < 60 mL/min/1.73 m^2^.

### 2.4. Echocardiography

Transthoracic echocardiography was performed in all participants to the study using a General Electric Vivid E9 (GE Health Medical, Horten, Norway) echocardiograph with a M5S 1.5/4.6 MHz active matrix-phased array transducer. Examinations were performed by two physicians (L.S. and C.O.), each with more than 10 years of experience in the field, and blinded to all clinical and laboratory data. Echocardiography data was collected and reported according to guidelines from the European Association of Echocardiography and the American Society of Echocardiography [19,20,22,23]. The severity of AS was quantified by continuous wave Doppler measurements of peak aortic flow velocity and mean transaortic gradients; AVA was calculated using the continuity equation and indexed to body surface area (BSA). LV systolic function was quantified by the LVEF (bi-plane Simpson’s modified rule). LV diastolic function and right chambers dimensions and functions were assessed following global recommendations.

### 2.5. Cardiac Magnetic Resonance Imaging

All CMR imaging examinations were performed by two experienced examiners, one cardiologist and one radiologist (L.A.C. and C.C.), blinded to all clinical data, using specialized software (Syngo.Via, VB20A_HF04, Argus, Siemens Healthineers Global, Erlangen, Germany), as recommended [24], on a 1.5 T MR scanner (Magnetom Symphony, Siemens Medical Solutions, Erlangen, Germany) with a dedicated surface coil for radio frequency signal detection. Standard localized views and contiguous short-axis and 4-chamber cine views covering both ventricles from base to apex were first acquired by ECG-gated steady-state free precession (SSFP) sequences, in apnea (Figure 2). Pre-contrast imaging parameters were selected at the beginning of the study and a standard protocol was applied for each examination: repetition time 3.6 ms; echo time 1.8 ms; flip angle 60°; slice thickness 6 mm; field of view 360 mm; image matrix of 192 × 192 pixels; voxel size 1.9 × 1.9 × 6 mm; 25–40 ms temporal resolution reconstructed to 25 cardiac phases. The imaging plane of the aortic valve was defined by the acquisition of a systolic three-chamber view and an oblique coronal view of the aortic valve and proximal aorta. Post-contrast, standard LGE images were acquired 10 minutes after intravenous injection of 0.2 mmol/kg gadolinium contrast agent (Gadoterate meglumine or Dotarem, Guerbet, Roissy CdG, France) in long- and short axis-views, using a segmented inversion-recovery gradient-echo sequence. Inversion time was adjusted to optimize nulling of apparently normal myocardium. Brachial blood pressure was monitored during CMR SSFP acquisitions. LV end-diastolic volume (LVEDV) and end-systolic volume (LVESV), LVEF and end-diastolic LV mass (LVM) were measured on short-axis cine-SSFP images. Epicardial and endocardial borders were traced semi-automatically at end-diastole and end-systole using specialized software (Syngo.Via VB20A_HF04, Argus, Siemens Medical Solutions). AVA was measured on cross-sectional planimetric images during systole by drawing the region of interest. LV longitudinal function was assessed by LAS, defined as the difference in mitral annular displacement at end-systole vs. end-diastole, and expressed in percentages. The presence and distribution of LGE in the LV were assessed from short-axis images, using the 17-segments model, recommended by the American Heart Association [24].The pattern of LGE distribution was characterized as mid-wall, subepicardial, focal or diffuse. 

### 2.6. Clinical Outcomes

Patients were followed-up over a median time interval of 386 days (interquartile range: 60 to 730 days) by completing a questionnaire either on hospital visits, telephone house-calls, or both. Survival analysis was performed for the combined end-point. The study end-point was a combination of major adverse cardiac events (MACE), including sudden cardiac death, non-fatal myocardial infarction, sustained ventricular arrhythmias, third-degree atrioventricular block and hospitalization for heart failure. If more than one event occurred, the most severe was selected. Hospitalizations due to non-cardiac causes were not considered in the analysis. The duration of follow-up was determined from the CMR study date until the occurrence of a MACE. The last evaluation of patient survival status was performed in August 2018 (the follow-up closing date). Complete follow-up was available for all patients.

### 2.7. Statistical Analysis

The normal distribution of data was tested by the Kolmogorov-Smirnoff test. Data were reported as mean and standard deviation, or median and interquartile range (IQR). Relative frequencies for categorical variables were reported as percentages. The Chi-square or Fisher’s exact tests were used to compare variables between the two groups. Multiple comparisons were made using one-way ANOVA variance analysis or the Kruskall–Wallis test. Univariable and stepwise multivariable logistic regression analyses were performed to determine the association between LAS and LGE, and other variables derived from imaging modalities. The hazard ratio (HR) for the prediction of events was calculated for each of the outcomes using a Cox regression model. For each outcome of interest, we considered all of the significant variables in the univariate analysis and sought the best overall multivariate models for the composite end point, by stepwise-forward selection, with a probability to enter set at *p* < 0.05 and to remove the effect from of regression at *p* < 0.05. Event-free survival (time to first event) was generated by the Kaplan–Meier method and statistical significance was determined by the log-rank test. Receiver operating characteristic curve analysis was performed to study the predictive ability of LAS and LVEF for adverse cardio-vascular events. Results were considered statistically significant for *p* < 0.05. Cohen’s Kappa inter- and intra-observer coefficient calculation was performed. Retrospective test power calculation and prospective sample size were estimated, with type I and type II variation according to sample size. The statistical analysis was performed using the MedCalc Software, version 16.1.2 (Mariakerke, Belgium).

## 3. Results

### 3.1. Baseline Characteristics

The present study was designed as a pilot study to serve as data for sample size calculation. Based on LAS numeric values and considering the threshold of alpha = 0.05, our estimation of type II beta risk stands below 0.05. However, in order to ensure the best power for inter-group tests, we have calculated a minimum necessary sample of 50 subjects /group (for type I alpha = 0.05). The study will be continued up to this limit for enabling us to decrease the type II error up to 0.05 (95% power).

All baseline characteristics of patients with AS and healthy volunteers are reported in Table 1. There were no statistically significant differences in age, gender, body surface area, systemic artery pressures, active smoking or presence of diabetes between the two groups. Patients in the test group had a poorer exercise capacity as assessed by the 6MWT.

Patients in the AS group had significantly higher levels of PICP (*p* < 0.001), PIIINP (*p* < 0.01), hs-CRP (*p* < 0.001) and NT-proBNP (*p* < 0.001) than patients in the control group. 

The cause of AS in test group patients was calcification of a trileaflet valve (*n* = 39), presence of a bicuspid aortic valve (*n* = 6), rheumatic disease (*n* = 4), or undetermined (*n* = 3). 47 (90%) patients with AS were symptomatic. LVEF was preserved in most test group patients (*n* = 38; 73%). Indexed LVM was increased ≥92 g/m^2^ in 28 patients (53.8%) and reduced LAS < −18% was documented in 20 patients with AS (38.4%). LGE was found in 30 patients with AS (57.7%). LGE was distributed mid-wall in 12 patients (23%), in the sub-epicardial myocardium in 5 patients (9.6%), was focal in 10 patients (19.2%), and diffuse in 3 patients (5.7%).

### 3.2. Reproducibility of CMR Measurements

CMR measurements were repeatedly performed on the same set of images, acquired from all patients in the study group. Intra- and inter-observer reproducibility of LVEF and LAS measurements, and the assessment of LGE by CMR were excellent. The kappa coefficients of agreement were 0.89 (inter-reader) and 0.91 (intra-reader) for the assessment of LGE, and 0.93 (inter-reader) and 0.96 (intra-reader) for LAS (Table 2).

### 3.3. Survival Analysis

All patients underwent surgery and aortic valve replacement was performed. During a median follow-up period of 386 (60 to 730) days, 22 patients (42.3%) had MACEs: non-fatal myocardial infarction (*n* = 2), sustained ventricular arrhythmias (*n* = 2), third-degree atrioventricular block (*n* = 3) and hospitalization for heart failure (*n* = 15). In three patients, MACEs (ventricular tachycardia and hospitalization for heart failure, respectively) occurred before surgery. One patient developed third degree atrio-ventricular block during surgery and required permanent pacing. Nineteen other patients experienced MACEs after aortic valve replacement. Most patients (*n* = 17, 77.2%) with LGE on CMR imaging had MACEs during follow-up.

The discriminative performance of additional markers for MACEs was calculated by receiver operating characteristic curve analysis for the combined end-point: LVEF (optimized cut-off value, <50%; AUC, 0.759; *p* < 0.01) and LAS (optimized cut-off value, <−18%; AUC, 0.883; *p* < 0.0001) (Table 3). Interestingly, the biomarkers for enhanced collagen turnover had no discriminative ability for MACEs in severe AS patients.

Kaplan–Meier curves for event-free survival showed a significantly higher rate of MACEs in patients with LGE (*p* < 0.01) and decreased LAS (*p* < 0.001) at CMR imaging (Figure 3).

### 3.4. Uni- and Multivariate Analysis

Several clinical parameters were predictors for the combined end-point in univariate analysis: the 6MWD, E/E’ ratio, LVEDV indexed, LVESV indexed, LVEF, LGE and LAS (Table 3). 

In multivariate Cox regression analysis, only LGE and LAS (adjusted HR = 9.86, 95% CI 1.77 to 54.0, *p* < 0.01, respectively, HR = 1.32, 95% CI 1.01–1.71, *p* < 0.01) remained independent predictors for MACEs (Table 4).

Subsequently, a stepwise multivariate Cox regression model was constructed, including age, 6MWD, E/E’ratio, LVEF, LAS and the presence of LGE. Only reduced LAS (HR 1.33 (95% CI 1.01–1.74; chi-square: 15.1, *p* < 0.001) and LGE (HR 11.3 (95% CI 1.82–70.2); chi-square: 24.3, *p* < 0.001) were independent predictors for the combined end-point (Table 5).

### 3.5. Incremental Predictive Value for the Combined End-Point

Sequential Cox proportional-hazards models yielded significantly increased predictive power for the combined end-point of MACEs when both LGE and LAS were used in addition to LVEF (chi-square = 19.74, one degree of freedom). Neither LGE, nor LAS provided incremental predictive power when used alone, in addition to LVEF (Figure 4).

## 4. Discussion

In this prospective pilot study, we aimed to establish whether advanced CMR imaging techniques provide additional value in identifying patients with severe AS at risk for MACEs. 

We have shown that both LAS and the presence of LGE are more reliable than LVEF for predicting MACEs in patients with severe AS, and that their combined use, in addition to LVEF, provided incremental value to any marker used alone. 

Earlier echocardiography-based studies have demonstrated that global longitudinal strain is a reliable marker of cardiovascular events in different categories of patients [25,26] and is superior to traditional measurements of LVEF for predicting MACEs in patients with cardiovascular diseases [27].

Two relatively recent studies confirmed that CMR feature tracking with SSFP is a robust method for assessing LV mechanics and compares well to global longitudinal strain assessed by echocardiography [28,29]. Also, studies have recently emerged showing that impaired myocardial strain at CMR identifies ventricular function impairment in preclinical arrhythmogenic right ventricular dysplasia [30] and light chain amyloidosis, even if LVEF is preserved [31]. Considering these findings, myocardial strain assessment by CMR might become a valuable tool for the early identification of myocardial impairment. 

Currently, data on the use of strain assessment by CMR in patients with AS is scarce. Al Musa et al. reported that strain by CMR is significantly decreased in patients with severe AS and preserved LVEF by comparison with healthy volunteers, and is even more impaired if patients are symptomatic [32]. To our knowledge, there is currently no other published data on the prognostic use of LAS in such patients, and our research is the first to address the relation with cardiovascular events. 

By contrast, the presence of LGE has already been validated by previous studies as a marker of poor prognosis in severe AS. The presence of LGE was associated with poorer outcomes after aortic valve replacement, including the lack of improvement in symptoms and increased mortality [12,33]. Moreover, in the study by Dweck et al., LGE provided incremental value to LVEF for survival in patients with moderate to severe AS [10]. The quantitative assessment of LGE also seems useful for predicting functional improvement and all-cause mortality after aortic valve replacement [10].

Although LGE assessment is obviously a valuable tool for risk stratification, in our study the predictive power for MACEs was increased when LAS was added to the model. By contrast, PICP and PIIINP did not contribute at all, which is consistent with previously published research [34]. Considering the results of our research and previously published data, LAS might become an important part of LV function assessment at CMR, particularly considering that it does not require contrast administration. Further research is needed to validate this hypothesis. 

Firstly, the study was conducted in a single institution and has the inherent limitations of that approach. Secondly, endomyocardial biopsies were not performed during surgery to assess the presence or absence of myocardial fibrosis in AS patients. Also, we were unable to acquire T1 mapping sequences, and therefore extracellular volume and diffuse myocardial fibrosis could not be quantified.

Finally, LAS, as a marker of longitudinal contractile function, is an independent predictor of outcomes in patients with AS and provides incremental value beyond the assessment of LVEF and the presence of LGE. 

We hypothesise that, if more data is collected to endorse it, LGE and LAS at CMR should provide incremental value to current risk scores for stratifying prognosis in patients undergoing surgical aortic valve replacement.

## Figures and Tables

**Figure 1 jcm-08-00165-f001:**
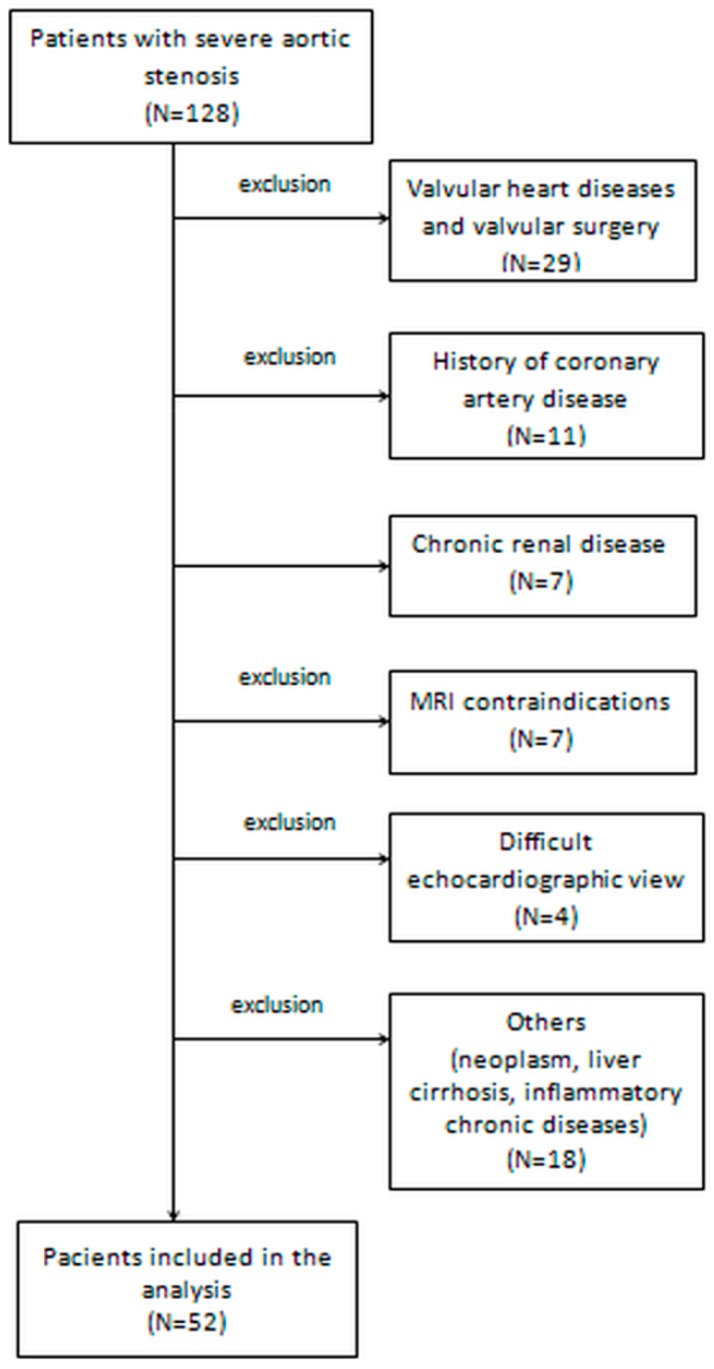
Patient selection and study design.

**Figure 2 jcm-08-00165-f002:**
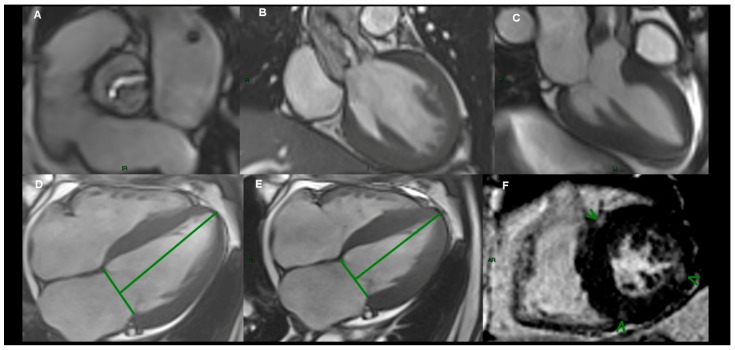
Cardiovascular magnetic resonance (CMR) imaging in aortic stenosis (AS) patients. (**A**) cine-SSFP (steady-state free precession) imaging of a stenotic bicuspid aortic valve; (**B**) coronal left ventricular outflow tract view acquired through-plane showing the aortic valve leaflet tips and restricted leaflet motility and resultant high velocity jet; (**C**) three-chamber view acquired through-plane showing the aortic valve leaflet tips, restricted leaflet mobility, and high velocity jet; (**D**,**E**) four-chamber view at end-diastole and end-systole acquired for longitudinal axis strain (LAS); (**F**) late gadolinium enhancement (LGE) in short-axis views of left ventricle showing focal hyper-enhancement (arrow).

**Figure 3 jcm-08-00165-f003:**
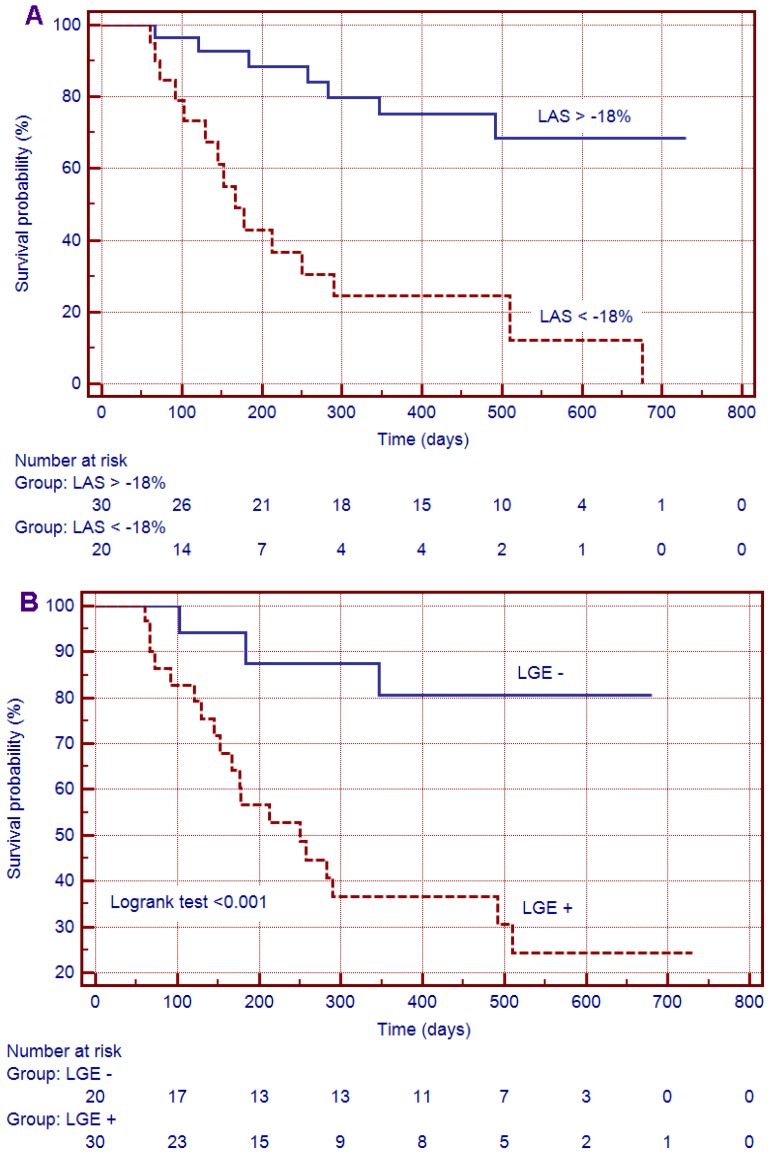
Kaplan–Meier curves for event-free survival for (**A**) Longitudinal Axis Strain (LAS); (**B**) late gadolinium enhancement (LGE).

**Figure 4 jcm-08-00165-f004:**
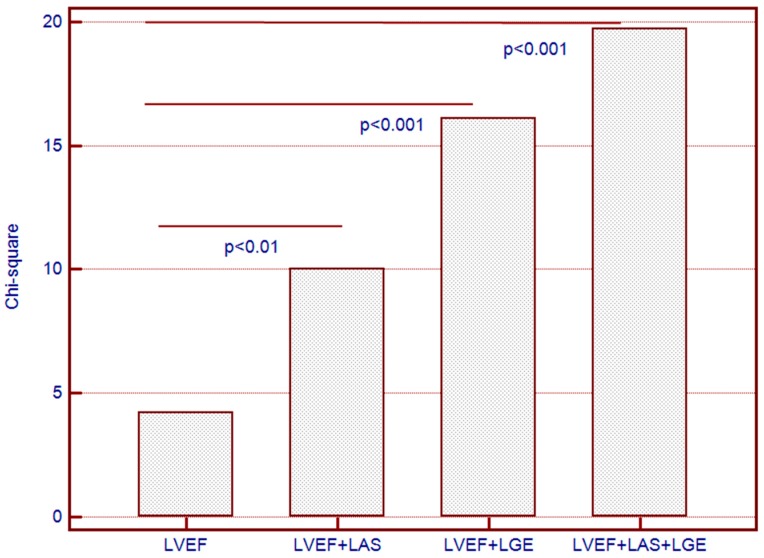
Incremental predictive value of longitudinal axis strain (LAS) and late gadolinium enhancement (LGE) added to left ventricular ejection fraction (LVEF) for the combined end-point in AS patients.

**Table 1 jcm-08-00165-t001:** Baseline characteristics of patients in the test and control group.

	Test Group	Control Group	*p*-Value
***Clinical characteristics***	*n* = 52	*n* = 52	
Age, years	66 (7.5)	66 (7.8)	NS
Male gender, *n* (%)	29 (55.7)	29 (55.7)	NS
Body surface area, m^2^	1.90 (0.24)	1.97 (0.13)	NS
Body-mass index, kg/m^2^	28.5 (4.1)	30.2 (4.9)	NS
Heart rate, bpm	73 (11.6)	72 (8.6)	NS
Systolic blood pressure, mmHg	132 (18.1)	133 (20.3)	NS
Hypertension, *n* (%)	37 (71.1)	28 (53.8)	NS
Diabetes mellitus, *n* (%)	22 (42.3)	14 (26.9)	<0.01
Dyslipidemia, *n* (%)	35 (67.3)	24 (46.1)	NS
Smoking, *n* (%)	19 (36.5)	13 (25)	NS
6MWD, m	406 (138.1)	592 (103.9)	<0.001
Coronary artery disease, *n* (%)	18 (32.6)		
Chronic obstructive lung disease, *n* (%)	7 (11.5)		
Peripheral vascular disease, *n* (%)	27 (51.9)		
NYHA functional class ≥ III, *n* (%)	15 (28.8)		
Logistic EuroScore, %	3.8 (1.3–5.9)		
***Medications***			
β-blockers, *n* (%)	40 (76.9)	14 (26.9)	<0.001
ACEIs or ARBs, *n* (%)	45 (86.5)	10 (19.2)	<0.001
Calcium channel blockers, *n* (%)	6 (11.5)	13 (25)	<0.01
Statins, *n* (%)	38 (73)	15 (28.8)	<0.001
ASA or other antiplatelet therapy, *n* (%)	32 (61.5)	13 (34.6)	<0.01
Diuretics, *n* (%)	37 (71.1)	5 (9.6)	<0.001
***Echocardiography***			
Peak aortic velocity, m/s	4.45 (0.47)	1.31 (0.36)	<0.001
Peak transaortic gradient, mmHg	82.1 (17.9)	7.2 (2.7)	<0.001
Mean transaortic gradient, mmHg	52.9 (14.7)	3.6 (0.75)	<0.001
AVA index, cm^2^/m^2^	0.52 (0.08)	2.9 (0.08)	<0.001
E/E’ ratio	9.8 (3.2)	6.5 (0.8)	<0.001
DT, ms	223 (52.2)	185 (8.8)	<0.001
sPAP, mmHg	33.4 (7.3)	26.2 (7.2)	NS
***Cardiovascular magnetic resonance***			
LVEDV index, mL/m^2^	82.4 (21.6)	61.8 (15.0)	<0.001
LVESV index, mL/m^2^	35.7 (16.6)	20.9 (5.8)	<0.001
LVM index, g/m^2^	96.2 (24.3)	62.1 (16.5)	<0.001
LVEF, %	58.4 (9.7)	66.1 (4.7)	<0.001
LVM/LVEDV, g/mL	1.22 (0.35)	1.04 (0.29)	<0.01
LAV index, mL/m^2^	49.1 (11.6)	25.5 (3.7)	<0.001
LAS (%)	−17.7 (3.9)	−20.5 (1.5)	<0.001
TAPSE, mm	14.9 (2.5)	19.8 (3.6)	<0.001
LGE, *n* (%)	30 (57.7)		
***Biomarker levels***			
PICP, ng/mL, IQR	1.2 (0.37–7.3)	0.42 (0.38–4.6)	<0.001
PIIINP, ng/mL, IQR	13.6 (2.5–68.3)	9.7 (2.4–29.7)	<0.01
hs-CRP, pg/mL, IQR	1.1 (0.49–1.9)	0.74 (0.16–1.1)	<0.001
NT-proBNP, pg/mL, IQR	1960 (170–9893)	210 (60–390)	<0.001
eGFR, ml/min/1.73 m^2^	88.1 (24.1)	89.2 (19.6)	NS

Abbreviations: *n*, number of patients; IQR, interquartile range; NYHA, New York Heart Association; NT-proBNP, N-terminal pro-Brain Natriuretic Peptide; hs-CRP, high sensitive C reactive protein; PICP, procollagen type I C-terminal propeptide; PIIINP, procollagen type III N-terminal propeptide; eGFR, estimated glomerular filtration rate; ACEI, angiotensin converting enzyme inhibitor; ARB, angiotensin receptor blocker; ASA, acetylsalicylic acid; LAS, left ventricular longitudinal-axis strain; LGE, left ventricular late gadolinium enhancement; LVEDV, left ventricular end-diastolic volume; LVESV, left ventricular end-systolic volume; LVM, left ventricular mass; LVEF, left ventricular ejection fraction; LAV, left atrial volume; E, peak mitral flow velocity; E’, early diastolic peak myocardial velocity; DT, early diastolic filling deceleration time; sPAP, systolic pulmonary artery pressure; 6MWD, six minute walk distance; TAPSE, tricuspid annular plane systolic excursion; AVA, aortic valve area. Data are reported as mean (standard deviation) or median (IQR) or *n* (%).

**Table 2 jcm-08-00165-t002:** Reproducibility inter and intra-reader agreement of CMR measurements.

Parameter	Coefficient Kappa	95% Confidence Interval	Standard Error
Inter-reader			
LVEF	0.95	0.907 to 0.974	0.023
LAS	0.93	0.912 to 0.962	0.027
LGE	0.89	0.795 to 0.940	0.078
Intra-reader			
LVEF	0.99	0.989 to 0.998	0.002
LAS	0.96	0.953 to 0.985	0.014
LGE	0.91	0.905 to 0.942	0.033

Abbreviations: LAS, left ventricular longitudinal-axis strain; LGE, left ventricular late gadolinium enhancement; LVEF, left ventricular ejection fraction.

**Table 3 jcm-08-00165-t003:** Predictive ability of biological markers and imaging parameters for outcomes in severe AS patients considered for aortic valve replacement surgery.

Variables	Sensibility	Specificity	PPV	NPV	ROC Threshold	AUC
LVEF	0.67	0.90	0.87	0.73	<50	0.759
LGE	0.75	0.68	0.66	0.76	+	0.717
LAS	0.77	0.90	0.85	0.84	<−18	0.883
PICP	0.56	0.73	0.64	0.67	>0.84	0.535
PIIINP	0.50	0.79	0.67	0.65	>16.1	0.572

Abbreviations: PICP, procollagen type I C-terminal propeptide; PIIINP, procollagen type III N-terminal propeptide; LAS, left ventricular longitudinal axis strain; LGE, left ventricular late gadolinium enhancement; LVEF, left ventricular ejection fraction.

**Table 4 jcm-08-00165-t004:** Univariate and Multivariate Cox Analysis testing between studied parameters and major cardiovascular events (MACEs).

	No Events *n* = 30	Events *n* = 22	Univariate Analysis		Multivariate Analysis	
			Unadjusted HR (95% CI)	*p* Value	Adjusted HR (95% CI)	*p* Value
Age, years	66 (10.1)	68 (7.1)	1.02 (0.95–1.09)	NS		
Male gender, *n*, %	14 (46.6)	15 (68.1)	0.40 (0.12–1.28)	NS		
Body surface area, m^2^	1.91 (0.27)	1.89 (0.20)	0.76 (0.08–7.20)	NS		
Systolic blood pressure	131 (10.5)	133 (15.2)	1.00 (0.97–1.03)	NS		
PICP, ng/mL, IQR	1.2 (0.37–5.0)	0.81 (0.38–7.3)	1.06 (0.76–1.49)	NS		
PIIINP, ng/mL, IQR	10.5 (6.4–68.3)	14.1 (2.5–56.8)	1.01 (0.97–1.06)	NS		
hs-CRP, pg/mL	1.1 (0.49–1.9)	0.94 (0.51–1.8)	0.18 (0.03–0.95)	NS		
NT-proBNP, pg/mL	2206 (170–6735)	2734 (234–9893)	1.00 (0.99–1.01)	NS		
eGFR, ml/min/1.73 m^2^	91.9 (25.4)	88.5 (22.6)	0.99 (0.97–1.01)	NS		
6MWD, m	455 (129)	340 (122)	0.99 (0.98–1.00)	0.001	0.99 (0.98–1.00)	NS
LVEDV index, mL/m^2^	75.3 (20.9)	92.1 (18.8)	1.02 (0.99–1.06)	<0.05		
LVESV index, mL/m^2^	29.9 (13.3)	43.6 (17.8)	1.04 (1.01–1.08)	<0.05		
LVM index, g/m^2^	93.2 (25.4)	100.3 (22.5)	1.01 (0.98–1.03)	NS		
LVEF, %	61.6 (7.9)	54.1 (10.5)	0.93 (0.87–0.99)	<0.01	0.97 (0.88–1.07)	NS
LAV index, mL/m^2^	49.8 (11.8)	48.2 (11.6)	0.98 (0.94–1.03)	NS		
LVM/LVEDV, g/mL	1.29 (0.36)	1.12 (0.31)	0.23 (0.04–1.27)	NS		
LGE, *n* (%)	12 (40)	17 (77.2)	5.55 (1.50–20.5)	<0.001	9.86 (1.77–54.0)	<0.01
LAS (%)	−19.6 (3.1)	−15.1 (3.3)	1.29 (1.07–1.55)	<0.001	1.32 (1.01–1.71)	<0.01
TAPSE, mm	19.3 (3.1)	20.4 (4.3)	1.08 (0.93–1.26)	NS		
E/E’ ratio	8.9 (1.9)	11.1 (4.1)	1.25 (1.02–1.53)	<0.01	1.36 (0.98–1.88)	NS
Peak aortic velocity, m/s	4.35 (0.33)	4.59 (0.59)	3.18 (0.84–11.9)	NS		
Peak aortic gradient, mmHg	78.7 (12.9)	86.9 (22.6)	1.02 (0.99–1.06)	NS		
Mean aortic gradient, mmHg	51.5 (12.5)	54.7 (17.4)	1.01 (0.97–1.05)	NS		
AVA index, cm^2^/m^2^	0.52 (0.08)	0.51 (0.08)	0.17 (0.08–0.98)	NS		

Abbreviations: *n*, number of patients; NT-proBNP, N-terminal pro-Brain Natriuretic Peptide; hs-CRP, high sensitive C reactive protein; eGFR, estimated glomerular filtration rate; PICP, procollagen type I C-terminalpropeptide; PIIINP, procollagen type III N-terminal propeptide; LAS, left ventricular longitudinal-axis strain; LGE, left ventricular late gadolinium enhancement; LVEDV, left ventricular end-diastolic volume; LVESV, left ventricular end-systolic volume; LVM, left ventricular mass; LVEF, left ventricular ejection fraction; LAV, left atrial volume; E, peak mitral flow velocity; E’, peak myocardial velocity at the mitral valve annulus; DT, early diastolic filling deceleration time; sPAP, systolic pulmonary artery pressure; TAPSE, tricuspid annular plane systolic excursion; AVA, aortic valve area; 6MWD, six minute walk distance.

**Table 5 jcm-08-00165-t005:** Stepwise Multivariate Proportional Hazard Model for the Combined End Point.

Variables	Model 1	Model 2	Model 3	Model 4	Model 5	Model 6
	HR 95%	HR 95%	HR 95%	HR 95%	HR 95%	HR 95%
Age	1.01 (0.95–1.08)	0.96 (0.89–1.03)	1.01 (0.95–1.08)	1.00 (0.95–1.06)	1.00 (0.95–1.06)	1.01 (0.96–1.07)
6MWD		0.99 (0.98–1.00)				
E/E’			1.24 (1.01–1.54) **			
LVEF				0.94 (0.88–1.01)		
LAS					1.33 (1.01–1.74) **	
LGE						11.3 (1.82–70.2) *

Abbreviations: 6MWD, six minute walk distance; E, peak mitral flow velocity; E’, peak myocardial velocity at the mitral valve annulus; LAS, left ventricular longitudinal-axis strain; LGE, left ventricular late gadolinium enhancement; LVEF, left ventricular ejection fraction. * *p* < 0.001; ** *p* < 0.01.

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
