# Peer review of "Incremental Predictive Value of Longitudinal Axis Strain and Late Gadolinium Enhancement Using Standard CMR Imaging in Patients with Aortic Stenosis"

_jcm, 2019, doi:10.3390/jcm8020165_

Reviewer 1 Report

Agoston-Coldea et al. present a prospective study performed in a cohort of patients with severe aortic stenosis. There was assessment of biomarkers of collagen turnover and extensive imaging evaluation (echocardiography and CMR). Main finding was that reduced global longitudinal strain assessed by CMR and the presence of LGE were strong independent predictors for MACE during follow-up. The study was well conducted and the methodology adequate. Nevertheless, findings are not very innovative nor really surprising. Several issues should be adressed:

- Were hemodynamic information available (e.g. by cath eaminations)? How was the correlation between hemodynamic data, imaging results and outcomes?

- The authors do not present information on procedural outcomes. This is somewhat remarkable, since there is a strong correlation of comlications and outcomes.

- LAS and LGE actually provide additional progostic information rather than really being independent. The reviewer would suggest to perform statistical tests such as integrated discrimination index and net reclassification index to prove their conclusions.

Author Response

Agoston-Coldea et al. present a prospective study performed in a cohort of patients with severe aortic stenosis. There was assessment of biomarkers of collagen turnover and extensive imaging evaluation (echocardiography and CMR). Main finding was that reduced global longitudinal strain assessed by CMR and the presence of LGE were strong independent predictors for MACE during follow-up. The study was well conducted and the methodology adequate. Nevertheless, findings are not very innovative nor really surprising. Several issues should be addressed:

- Were hemodynamic information available (e.g. by cath examinations)? How was the correlation between hemodynamic data, imaging results and outcomes?

Hemodynamic data was not available, as cath examinations other than coronary angiography were not performed in all cases.

- The authors do not present information on procedural outcomes. This is somewhat remarkable, since there is a strong correlation of complications and outcomes.

We added that all patients underwent surgery for aortic valve replacement and mentioned that one patient developed third atrio-ventricular block during surgery, which required permanent pacing.

- LAS and LGE actually provide additional prognostic information rather than really being independent. The reviewer would suggest to perform statistical tests such as integrated discrimination index and net reclassification index to prove their conclusions.

We thank the reviewer for the suggestions. We have, however, considered models for NRI as bringing no stronger support for our conclusions. Recent literature brings evidence regarding their limitations, especially due to arbitrary selection of thresholds (Pickering, John & Endre, Zoltan. (2012). New Metrics for Assessing Diagnostic Potential of Candidate Biomarkers. Clinical journal of the American Society of Nephrology : CJASN. 7. 1355-64. 10.2215/CJN.09590911.)According to the same study, which employed risk assessment for detecting differences between statistical metric systems, the IDI could present a better  performance, but only when applied separately on the control and test group. Therefore,  assuring  a proper index power would completely shift the direction of analyzes from the aim of the study.

Reviewer 2 Report

In this prospective study the authors analysed the value of Cardiovascular magnetic resonance (CMR) late gadolinium enhancement (LGE) imaging and left ventricular (LV) long-axis strain (LAS) assessment in predicting the prognosis in patients with severe aortic valve stenosis.

52 patients with severe aortic stenosis who had an indication for surgery and 52 healthy volunteers were included. Beside clinical examination, 12-lead ECG, 24-hour Holter monitoring, 6-minute walk test, echocardiography, biochemical analysis and echocardiography, patients underwent CMR imaging. CMR imaging data were used for LV volumetric assessment, assessment of myocardial fibrosis (LGE) as well as measurement of the aortic valve area and LV long-axis strain (LAS). Median follow-up period was 386 days and during that period 22 major cardiovascular events (MACE) occurred.

Kaplan-Meier survival curves for event-free survival demonstrated a higher rate of MACE in aortic stenosis patients with LGE and decreased LAS values. Cox regression analysis showed that LGE and LAS were independent predictors of MACE. The predictive value of LGE and LAS increased when they were used combined, in addition to LVEF. 

The study is interesting, and the results might be of clinical use, however, no surgical information is provided. Could any of the events be related to the surgical procedure? This needs to be clarified.

Specific comments:

1.     In the “Material and Methods” section it says that patients with rheumatic aortic stenosis were excluded. However, in the result section (page 9) it says “The cause of AS in test group patients was calcification of a trileaflet valve (n=39), presence of a 222 bicuspid aortic valve (n=6), rheumatic disease (n=4), or undetermined (n=3).”

I suggest that the authors comment on this. 

2.     The authors state that controls with a “history of chronic heart disease, other than systemic arterial hypertension” were excluded. Why were controls with arterial hypertension and diabetes mellitus not excluded from this study (table 1)?

3.     Do the authors have performed CMR feature tracking? If so, what were the results?

4.     Reproducibility of CMR measurements (page 9): I suggest including a table showing the results.

5.     Survival analysis (page 9): The authors note that “During a median follow-up period of 386 (60 to 730) days, 22 patients (42.3%) had MACEs: non-fatal myocardial infarction (n=2), sustained ventricular arrhythmias (n=2), third-degree atrioventricular block (n=3) and hospitalization for heart failure (n=14)”. However, the number of patients in brackets altogether is only 21, not 22. The authors should either correct the numbers or provide an explanation. 

6.     It remains unclear for the reader how many patients underwent aortic valve surgery. Furthermore, no details about surgery are provided (type of surgery). Were there any patients who had a TAVI procedure? I recommend including more details.

7.     Were any of the MACEs related to surgery?

8.     The authors write in the abstract and on page 9 (lines 239 & 240) that “Most patients (n= 17, 77.2%) with LGE on CMR imaging had MACEs during follow-up.”  However, I assume the authors mean that most patients who had a MACE were found to have LGE on CMR. Please correct the sentence.

9.     The discussion is short and should be more detailed. What are the advantages of CMR compared to echocardiography in this context? What do the authors suggest should be the clinical implications?

Minor comments:

1.     Abstract: Please mention that only 52 patients with aortic valve stenosis were included in this study. For the reader it seems as if 128 patients were included. 

2. Page 5, line 173: I assume the authors mean questionnaire instead of “query”?

Author Response

1.    In the “Material and Methods” section it says that patients with rheumatic aortic stenosis were excluded. However, in the result section (page 9) it says “The cause of AS in test group patients was calcification of a trileaflet valve (n=39), presence of a 222 bicuspid aortic valve (n=6), rheumatic disease (n=4), or undetermined (n=3).” I suggest that the authors comment on this.

We excluded patients with significant mitral stenosis due to rheumatism. We corrected the text accordingly.

2.      The authors state that controls with a “history of chronic heart disease, other than systemic arterial hypertension” were excluded. Why were controls with arterial hypertension and diabetes mellitus not excluded from this study (table 1)?

The patients in the aortic stenosis group were elderly – mean age 66 (±7.5) years old – and we could not find perfectly healthy controls of the same age.

3.     Do the authors have performed CMR feature tracking? If so, what were the results?

 CMR feature tracking was not performed because the adequate software was not available.

4.     Reproducibility of CMR measurements (page 9): I suggest including a table showing the results.

 The table was added as requested.

 Table 2. Reproducibility inter and intra-reader agreement of CMR measurements

Parameter

Coefficient kappa

95%   Confidence interval

Standard   error

Inter-reader

LVEF

0.95

0.907 to 0.974

0.023

LAS

0.93

0.912 to 0.962

0.027

LGE

0.89

0.795 to 0.940

0.078

Intra-reader

LVEF

0.99

0.989 to 0.998

0.002

LAS

0.96

0.953 to 0.985

0.014

LGE

0.91

0.905 to 0.942

0.033

Abbreviations: LAS, left ventricular longitudinal-axis strain; LGE, left ventricular late gadolinium enhancement; LVEF, left ventricular ejection fraction.

5.     Survival analysis (page 9): The authors note that “During a median follow-up period of 386 (60 to 730) days, 22 patients (42.3%) had MACEs: non-fatal myocardial infarction (n=2), sustained ventricular arrhythmias (n=2), third-degree atrioventricular block (n=3) and hospitalization for heart failure (n=14)”. However, the number of patients in brackets altogether is only 21, not 22. The authors should either correct the numbers or provide an explanation. 

There were 15 patients who were hospitalized for heart failure. We corrected the number in the manuscript.

6.     It remains unclear for the reader how many patients underwent aortic valve surgery. Furthermore, no details about surgery are provided (type of surgery). Were there any patients who had a TAVI procedure? I recommend including more details.

All patients underwent surgery for aortic valve replacement. TAVI was not performed. The information was added o the text (page 9, rows 236, 240-241).

7.     Were any of the MACEs related to surgery?

One patient developed third degree atrio-ventricular block during surgery and required permanent pacing.

8.     The authors write in the abstract and on page 9 (lines 239 & 240) that “Most patients (n= 17, 77.2%) with LGE on CMR imaging had MACEs during follow-up.”  However, I assume the authors mean that most patients who had a MACE were found to have LGE on CMR. Please correct the sentence.

 We meant to say that among the patients who had LGE on CMR, 17 (77.2%) had MACEs, not the other way around.

9.     The discussion is short and should be more detailed. What are the advantages of CMR compared to echocardiography in this context? What do the authors suggest should be the clinical implications?

We hypothesise that, if more data is collected to endorse it, LGE and LAS at CMR should provide incremental value to current risk scores for stratifying prognosis in patients undergoing surgical aortic valve replacement.

This hypothesis was included in the conclusion to express the clinical value that LGE and LAS at CMR might have in the future.

Minor comments:

    1.     Abstract: Please mention that only 52 patients with aortic valve stenosis were included in this study. For the reader it seems as if 128 patients were included. 

We corrected the number in the abstract.

2.      Page 5, line 173: I assume the authors mean questionnaire instead of “query”?

We replaced query by questionnaire in both the abstract and on page 5.

Round  2

Reviewer 1 Report

Responses are sufficient.

Reviewer 2 Report

The authors have satisfactorily responded to the reviewer comments. There is no need for further revision.